# The Role of Probiotics and Their Metabolites in the Treatment of Depression

**DOI:** 10.3390/molecules28073213

**Published:** 2023-04-04

**Authors:** Monika Elżbieta Jach, Anna Serefko, Aleksandra Szopa, Ewa Sajnaga, Hieronim Golczyk, Leandro Soares Santos, Kinga Borowicz-Reutt, Elwira Sieniawska

**Affiliations:** 1Department of Molecular Biology, The John Paul II Catholic University of Lublin, Konstantynów Street 1I, 20-708 Lublin, Poland; 2Department of Clinical Pharmacy and Pharmaceutical Care, Medical University of Lublin, Chodźki Street 1, 20-093 Lublin, Poland; 3Department of Biomedicine and Environmental Research, The John Paul II Catholic University of Lublin, Konstantynów Street 1J, 20-708 Lublin, Poland; 4Department of Animal and Rural Technology, State University of Southwest Bahia, Itapetinga 45700-000, BA, Brazil; 5Independent Unit of Experimental Neuropathophysiology, Department of Toxicology, Medical University of Lublin, Jaczewskiego 8b, 20-090 Lublin, Poland; kingaborowicz@umlub.pl; 6Department of Natural Products Chemistry, Medical University of Lublin, Chodźki Street 1, 20-093 Lublin, Poland

**Keywords:** gut–brain axis, depression, dysbiosis, microbiota, microbiota metabolites, metabiotics, probiotics, psychobiotics, prebiotics, synbiotics

## Abstract

Depression is a common and complex mental and emotional disorder that causes disability, morbidity, and quite often mortality around the world. Depression is closely related to several physical and metabolic conditions causing metabolic depression. Studies have indicated that there is a relationship between the intestinal microbiota and the brain, known as the gut–brain axis. While this microbiota–gut–brain connection is disturbed, dysfunctions of the brain, immune system, endocrine system, and gastrointestinal tract occur. Numerous studies show that intestinal dysbiosis characterized by abnormal microbiota and dysfunction of the microbiota–gut–brain axis could be a direct cause of mental and emotional disorders. Traditional treatment of depression includes psychotherapy and pharmacotherapy, and it mainly targets the brain. However, restoration of the intestinal microbiota and functions of the gut–brain axis via using probiotics, their metabolites, prebiotics, and healthy diet may alleviate depressive symptoms. Administration of probiotics labeled as psychobiotics and their metabolites as metabiotics, especially as an adjuvant to antidepressants, improves mental disorders. It is a new approach to the prevention, management, and treatment of mental and emotional illnesses, particularly major depressive disorder and metabolic depression. For the effectiveness of antidepressant therapy, psychobiotics should be administered at a dose higher than 1 billion CFU/day for at least 8 weeks.

## 1. Introduction

Depression is a common and complex mental disorder with approximately 350 million people affected globally [1,2,3]. Depression rates are predicted to increase due to the coronavirus pandemic and post-pandemic situation, particularly due to a growth in the incidence rate among younger individuals [1,4]. Depressive disorders currently impact more than 17.3 million adult Americans; however, it is observed that coefficients have tripled as a result of the stress caused by the COVID-19 pandemic [5]. Depression may have various psychological symptoms, from usual mood fluctuations and short-lived emotional responses to the feeling of hopelessness and helplessness, having no motivation or interest in things, experiencing no enjoyment of life, and having problems with everyday routines [1]. They last for at least two weeks [6,7]. Depending on the number and severity of symptoms, an episode of depression may be categorized as mild, moderate, or severe. Depressive patients can also be asymptomatic. There is also a significant difference between depression in individuals who have had or have not had a history of manic episodes. Both types of depression can be chronic with relapses, particularly if they are untreated [1,8]. Depression becomes a serious health condition when it lasts for a long time with moderate or severe intensity. This health issue is called major depressive disorder (MDD), which is also commonly known as clinical depression [1]. Among people with MDD, depressive episodes tend to recur with greater severity and less responsiveness to conventional treatments [8]. MDD negatively affects patients’ performance at school, work, and at home. Thus, it significantly diminishes their quality of life. Consequently, MDD often leads to suicidal attempts. It has been estimated that nearly 800 thousand individuals die due to suicide every year. Suicide is the second leading cause of death among young people between 15 and 29 years old [1]. 

In bipolar affective disorder, patients experience extreme mood swings that include emotional highs (mania or hypomania) and lows (depression). Manic episodes involve an elevated or irritable mood, over-activity, pressure of speech, inflated self-esteem, and a decreased need for sleep. Some mental and physical conditions can worsen symptoms of bipolar disorder or can make treatment less successful, including anxiety, attention-deficit/hyperactivity disorder (ADHD), alcohol or drug problems, eating disorders, heart diseases, headaches, obesity, or thyroid problems [9]. It has been shown that depressive disorders are two or three times more likely to develop in people that suffer from several other diseases than in people with a single physical disease or in those who have no chronic physical problems [10]. Depressive people may suffer from many metabolic conditions, e.g., they are diagnosed with metabolic syndrome (MetS) 1.5 times more often than non-depressive subjects [11,12]. MetS increases cardiovascular diseases and type 2 diabetes mellitus (T2D), which increases the mortality risk [13,14]. Additionally, a relationship between obesity and MetS with MDD has been observed [15,16,17]. Other conditions closely associated with depression and MetS are non-alcoholic fatty liver disease (NAFLD), also known as metabolic-associated fatty liver disease (MAFLD), i.e., a multi-complex health problem characterized by accumulation of fat in the liver [18,19,20], and cancer [21]. Consequently, in 2021, Gawlik-Kotelnicka and Strzelecki proposed the term “metabolic depression” to define depression comorbid with metabolic conditions [22]. 

There are several approved treatments for moderate and severe depression, including psychological therapy (such as behavioral activation, cognitive behavioral therapy, meditation, and interpersonal psychotherapy), pharmacotherapy (with antidepressant drugs, such as selective serotonin reuptake inhibitors, serotonin–norepinephrine reuptake inhibitors, atypical antidepressants, tricyclic antidepressants, serotonin antagonist and reuptake inhibitors, or lithium) [2], electroconvulsive therapy, deep brain stimulation, and bright light therapy. Antidepressants can be an effective form of treatment in moderate–severe depression; however, they can be problematic because of weight gain. A cohort study with 294,719 people showed that antidepressant drugs can contribute to long-term increased risk of ≥5% weight gain at the population level. It was noted that people who were initially of normal weight had an adjusted rate of transition to overweight or obesity of about 1.29; those who were initially overweight had an adjusted rate of transition to obesity of approximately 1.29 [23]. Therefore, gaining weight may increase depressive conditions. Furthermore, the current treatment for depression is insufficient since 30% of patients are treatment-resistant. Ketamine is a new and effective antidepressant in treatment-resistant individuals. It is recognized that the antidepressant impact of ketamine might be partially mediated by the modification of gut microbiota [24]. However, antidepressants are not the first-line treatment in cases of mild depression. Psychosocial treatments are preferred in this condition [2]. In addition to the above-mentioned therapies, natural or alternative therapy, such as herbal medicines, physical activity and exercise, meditation, mindfulness, nature therapy, and music therapy are used. Furthermore, a balanced diet and proper nutrition, avoiding stress, as well as getting plenty of sleep, are recommended [25,26,27,28,29,30]. People who are depressed are more likely to have lower levels of vitamins B12 and D; therefore, their supplementation could be particularly useful in the alleviation of depressive symptoms [31,32]. A novel approach to depression therapy, especially to the treatment of metabolic depression, proposes administration of probiotics [2,22,25,33,34] as a monotherapy or as an add-on to standard antidepressant therapy (TAU) [2,35,36]. The effectiveness of probiotics has been noted in other mental conditions, such as anxiety [37,38], fatigue, mood and sleep disturbances [34], stress and memory problems [38], anger [34,37], post-partum depression [39], anxiety, and depressive symptoms in schizophrenia [40]. 

Recently, it has been suggested that an association between the gut microbiota and the brain exists [25]. The intestinal microbiota connects the gastrointestinal tract (GIT) with the central nervous system (CNS) via a biochemical signaling pathway, including modulation of the availability of circulating serotonin, kynurenine, tryptophan, and short-chain fatty acids (SCFAs). The gut microbiota also influences the blood–brain barrier and its permeability, activation of peripheral immune system cells, and function of the brain microglia [41]. It has been observed that alterations in the bowel microbiota, called intestinal dysbiosis, occur in patients with chronic diseases, including MetS and depressive disorders [42,43,44,45,46]. When the gut microbiota composition is compromised, properties of the protective barrier are impaired, resulting in an increase in the permeability of the intestinal walls and, as a consequence, in the trespass of antigens into the bloodstream, which cause inflammation [47]. The penetration of the CNS by various substances may change the physiological functions of the brain [41]. It seems that the gut microbiota influences the brain primarily through the vagus nerve by humoral and neural means of the gut–brain axis [46]. Thus, intestinal dysbiosis leads to activation of the innate immune response, resulting in the chronic low-grade inflammation manifested by constantly high concentrations of pro-inflammatory mediators, and the loss of immune-regulatory function [48,49]. It has been widely proven that inflammation and neuropsychiatric diseases are closely related [50]. Hence, neuroinflammation is associated with activation of microglial cells and manifestation of peripheral infiltrating leucocytes in the CNS parenchyma [51]. In homeostasis, microglial cells do not produce pro-inflammatory agents, and neuroinflammation does not occur [41].

The gut microbiota is able to affect the physiological, behavioral, and cognitive functions of the brain [52]. Thus, probiotics and their metabolites may influence the treatment of depression [2,53]. Moreover, it is known that a well-balanced diet plays a crucial role in the management of acute and chronic diseases [53,54]. This paper aims to highlight a connection between mental and digestive health, the importance of the microbiota–gut–brain axis in depression, and the potential benefits of probiotics as psychobiotics in the prevention, management, and treatment of depressive disorders, especially in metabolic depression. We also analyzed the metabolites as metabiotics (postbiotics) responsible for the psychobiotic properties of specific probiotics. Furthermore, in this review, we summarized clinical data from the last five years in which the impact of different probiotics supplementations on depressive symptoms was assessed.

## 2. Intestinal Microbiota

The intestinal microbiota of the healthy individual consists of a compact and diverse community of microorganisms that derive from approximately 5000 various species and with over 7000 strains [55,56]. In total, 99% of these species represent the bacterial phyla: *Firmicutes* (Gram-positive), *Bacteroidetes* (Gram-negative), *Actinobacteria* (Gram-positive), and *Proteobacteria* (Gram-negative) [55]. However, approximately 90% of all bacterial species inhabiting the adult GIT belong to the core *Firmicutes* phylum (including the genera *Lactobacillus*, *Coprococcus*, *Clostridium*, and *Enterococcus*), which accounts for about 40–65% of the colon or fecal microbiota, and the *Bacteroidetes* phylum, collectively referred to as *Cytophaga–Flavobacterium–Bacteroides* (CFB) (e.g., *Bacteroides*, *Prevotella*, and *Desulfuribacillus* genus) [57,58,59,60]. Therefore, most of the beneficial bacteria that constitute the human intestinal microbiota are represented by *Firmicutes*, which are divided into the *Clostridium coccoides* (*Clostridium* cluster XIVa) and *Clostridium leptum* (Clostridium cluster IV) sub-groups, whereas the CFB group is mainly represented by a large number of *Prevotella* and *Porphyromonas* [60]. Furthermore, the low-abundance *Verrucomicrobia* phylum (e.g., *Akkermansia muciniphila*) determines potential benefits for human metabolism [61]. The gut microbiota also includes viruses, especially bacteriophages, Archaea, and Eukarya such as Fungi, Blastocystis, and Amoebozoa [60].

*Firmicutes* and *Bacteroidetes* dominate in the adult healthy GIT. However, switches in diet, as well as advancing age, cause changes in the number of individual genera. For example, an increase in the abundance of the bile-tolerant species from the *Bacteroidetes* phylum belonging to such genera as *Bacteroides* and *Alistipes*, and *Bilophila* from the *Proteobacteria*, and a decrease in *Firmicutes* are associated with protein- and fat-enriched diet [62]. In turn, the gut microbiota of the elderly is characterized by a decreased *Bacteroides* to *Firmicutes* ratio, and most of all, reduced *Bifidobacterium* abundance, amylolytic activity, and SCFA production as well as an increase in the abundance of Gram-negative rods from *Enterobacteriaceae*. Noteworthy, depression syndrome is one of the most common mood disorders in the late-life population [63].

The GIT ecosystem is assumed to be composed of more than 100 trillion microbial cells [64] with a mass weight of about 1–2 kg [65]. The microbiota lives predominantly in the small and large intestines [65]. In general, microbial density, measured as the colony forming unit (CFU), increases numerically from the duodenum (10^1–3^ CFU/mL) up to their maximum in the colon (10^11–12^ CFU/mL), and finally some cells of the gut microbiota leave the GIT with feces in a quantity of 10^10^ CFU/mL [66]. Such a quantitative distribution of gut bacteria is very important, and it is related to their metabolic activity. The processes of degradation of simple carbohydrates mainly take place in the small intestine [67]. In turn, complex carbohydrates largely are catabolized in the colon [68]. Through degradation of complex carbohydrates, and soluble and insoluble fiber from prebiotics, or protein and peptides from dietary proteins, many bacteria involved in the beneficial symbiotic relationship with a host human organism produce host-advantageous compounds with neuroactive properties such as SCFAs, especially acetate, butyrate, and propionate (e.g., *Akkermansia muciniphila*, *Bifidobacterium* spp.) [41,66,69], and tryptophan metabolites (e.g., *Bifidobacterium adolescents*, *Bifidobacterium longum*, *Bifidobacterium pseudolongum*, *Lactobacillus acidophilus*, *Lactobacillus reuteri*), including indole-3-acetic acid, indole-3-acetaldehyde, indole-3-propionic acid, indole-aldehyde, and indole acrylic acid [70]. Moreover, bacteria (e.g., consortia of *Bifidobacteria* spp.) help with the digestion of various substrates, e.g., indigestible plant biomass such as cellulose [71]. 

### 2.1. Metabiotics and Their Functions

Metabolites, also called metabiotics (postbiotics), produced by the intestinal microbiota engaged in the mutualistic or commensal relationship with their host, play a key role in host metabolisms and health, especially in the anti-inflammatory activity, as presented in Table 1. Such SCFAs as acetate and propionate are endogenous ligands of two G-protein-coupled receptors (GPR41 and GPR43). These ligands can modulate inflammation and enhance the production of glucagon-like peptide-1 and peptide tyrosine–tyrosine, which influence satiety and the GIT transit [72]. In turn, butyrate is the main energy source for colonocytes and protects against colorectal cancer and inflammation by inhibiting histone deacetylases (HDACs) [73]. All SCFAs have an inhibitory effect on HDACs; however, only butyrate influences specific short-chain fatty acid receptors/free fatty acid receptors (FFARs) such as GPR41/FFAR3, GPR43/FFAR2, and hydroxycarboxylic acid receptor GPR109A/HCAR2, which may inhibit the production of inflammatory cytokines. GPR109A/HCAR2 plays a significant role in regulating the blood–retinal barrier’s integrity and has therapeutic potential toward preventing and treating retinal diseases such as diabetic retinopathy, in which the compromised barrier function is of paramount importance [74]. 

Moreover, SCFAs regulate the blood–brain barrier’s permeability. In a preclinical study, an intraperitoneal and intravenous administration of sodium butyrate protected the blood–brain barrier from destruction and promoted neurogenesis and angiogenesis [91,92]. Sodium butyrate also alleviated the depressive-like behavior induced by lipopolysaccharides (LPS) and abolished hippocampal microglial activation in mice [93]. Furthermore, germ-free mice showed microglial defects with changed cell proportions leading to weakened innate immune responses. Recolonization of the gut microbiota restored microglial properties, demonstrating that SCFAs also regulate microglial homeostasis [75]. In an animal model of depressive-like behavior, administration of the probiotic metabolite resulted in a reduction in depressive-like behavior, raising prefrontal cortical ten-eleven translocation methylcytosine dioxygenase-1 and augmenting the brain-derived neurotrophic factor (BDNF) concentration [94]. In an autism spectrum disorder (ASD) mouse model, butyrate treatment improved long-term memory, hippocampal CA1 dendric spine density, and histone acetylation [95]. These results were confirmed in a clinical trial. In an Alzheimer’s clinical trial, butyrate treatment improved contextual fear learning, spatial learning, hippocampal synaptophysin synthesis, and neural plasticity [96].

Another SCFA, lactate was previously considered as a metabolic waste, but recent research has shown that, in a low concentration, it increases the transcription and protein levels of BDNF in neurons and microglial cells. In an animal model, lactate produced by astrocyte aerobic glycolysis played a crucial role in memory acquisition in mice and maintained learning-dependent synaptic plasticity, especially after moderate- to high-intensity exercise [82]. Noteworthy, intestinal communities with small numbers of lactate-utilizing bacteria in other SCFAs are definitely less stable and more inclined toward lactate-induced perturbations [83]. This confirms that the health of the host depends on the right proportion of specific genera and species of microorganisms. 

In turn, tryptophan metabolites (Table 1) produced by the gut microbiota from protein and peptide are mostly agonists of the aryl hydrocarbon receptor (AhR), which can overpass the blood–brain barrier. Tryptophan metabolites activate the AhR in astrocytes and in brain and spinal cord microglial cells, acting as a predominant source of immune defense in the CNS [51]. Therefore, activation of the AhR inhibits the transcription factor, e.g., nuclear factor-ĸB (NF-ĸB), blocking the production of pro-inflammatory cytokines and chemokines. The AhR activation also stimulates the expression of microglial transforming growth factor α (TGF-α), which affects astrocytes and abolishes their pro-inflammatory activity [97]. Moreover, the AhR activation directly enhances the expression of the cytokine signaling 2 inhibitor and indirectly stops the NF-ĸB signaling pathway [98]. Interestingly, a reduction in the production of the AhR ligands is observed in the gut microbiota collected from individuals with inflammatory bowel disease (IBD) [85]. In a proteomic research work, a postmortem comparison of the hippocampus from patients with schizophrenia and bipolar disease versus from healthy people showed that the hippocampus of schizophrenic individuals had significant abnormalities in the AhR signaling and in 14-3-3 proteins that affected many brain functions, especially neural signaling, neuronal development, and neuroprotection. By contrast, the hippocampal tissue collected from patients with bipolar disease showed distinct changes in glucose metabolism [98]. In another clinical trial, it was observed that ASD severity was associated with AhR-related gene variants [99].

### 2.2. Microbial Colonization and Microbiota–Host Relations

Microbial colonization of the human body, particularly GIT, begins at birth. An infant’s bowels are considered sterile or to contain a very low number of microorganisms [100]. During a vaginal birth, an infant is exposed to vaginal microbiota of the mother, and it is rapidly colonized by them [25,100]. Infants delivered by vaginal birth have a greater yield and diversity of microbial composition than the ones delivered by a Caesarean section [101,102]. The composition of the gut microbiota in newborn babies is unique. In the early years of life, the gastrointestinal microbiota in infants plays a significant role in maturation of the immune and intestinal systems and in protection against pathogens [101]. Breastfeeding also alters the gut microbiota in infants [100]. The microbiota of breast-fed babies is more abundant than that in mixed-fed or milk-powder-fed babies, particularly in relation to *Bifidobacterium* and genera from the *Lactobacillaceae* family [102,103], which have proven probiotic properties shown in Table 2.

Over the human life course, symbiotic and commensal microbiota with probiotic properties (Table 2), particularly lactic acid bacteria (LAB), lower pH in the gut’s environment due to the production of acids (particularly lactic acid but also acetic and propionic acids and several others) [103,112]. Furthermore, probiotic microbiota helps to reduce the growth of potentially pathogenic and obligatory parasitic microorganisms that inhabit the host GIT in negligible amounts (e.g., *Actinomycetes odontolyticus*, *Bdellovibrio* spp., *Clostridium difficile*, or *Candida albicans*), as well as limit gastrointestinal discomfort, bloating, and flatulence, and improve digestive regularity [113,114]. Moreover, microbiota with probiotic properties also improve skin functions and augment resistance to allergens, as well as protect DNA, proteins, and lipids from oxidative damage. They restore the gut microbiota in individuals treated with antibiotics [2,115,116,117,118]. 

### 2.3. Mechanisms of Microbiota in Gut–Brain Axis

It has been observed that the microbiota has a great influence on the mental state by the bidirectional communication between the gut and the brain [119,120,121]. The intestinal microbiota, being in symbiotic, commensal, and parasitic relationships with its host, can activate the CNS and the immune system. The gut microbiota is responsible for production of neuroactive substances, such as serotonin and gamma-aminobutyric acid (GABA) [122,123]. Furthermore, the intestinal microbiota contributes to the maintenance of homeostasis since it modulates pro- and anti-inflammatory responses [106]. These substances and mediators are transited by the vagus nerve as the most direct intermediatory pathway between the gut and the brain (80% afferent and 20% efferent fibers). Afferent fibers in the vagus nerve do not overpass the epithelial layer of the intestinal walls and are not in direct communication with the intestinal luminal microbiota [124]. Enteroendocrine cells cooperate with vagal afferents, either directly via serotonin release, activating 5-hydroxytryptamine-3 receptors in vagal afferent fibers, or indirectly via intestinal hormones. Interestingly, in the enteroendocrine cells, FFARs and toll-like receptors (TLRs) such as TLR1, TLR2, and TLR4 are present, which recognize elements of cell walls, locomotor system, single- and double-stranded RNA and DNA motifs typical of microorganisms [125]. Moreover, enteroendocrine cells detect endotoxic LPS of Gram-negative bacteria via TLR4 and peptidoglycan of Gram-positive bacteria via TLR2. Consequently, enteroendocrine cells identify bacterial compounds and exert an indirect effect on vagal afferent fibers by modulating GIT motility, secretion, and food intake. Therefore, the gut microbiota can interact with the vagus nerve via direct mechanisms, including TLRs activated by microbial elements, which additionally directly activate vagal afferent fibers at the nodose ganglia [41,124]. 

In the GIT, there are huge numbers of immune system cells, which are in constant contact with the gut microbiota [55]. Epithelial goblet cells secrete protective viscous mucins, forming the mucus layer, where the host–gut microbiota interaction occurs. The intestinal immune system maintains tolerance to commensal microbiota and resistance to pathogens. The imbalance between the host immune system and microbiota leads to inflammation and several diseases [41]. Many microbial compounds and elements, e.g., LPS, peptidoglycan, lipoteichoic acid, flagellum, pilus, DNA, and cell wall fragments, are recognized as pathogen-associated molecular patterns (PAMP) [41,126]. Receptors of pathogen recognition and non-pathogen recognition are responsible for the PAMP identification by immune cells [127]. Sensing the PAMP by the immune receptors launches a cascade of signaling pathways that activate many transcription factors and promote the production of pro-inflammatory agents such as antimicrobial peptides, cytokines, and chemokines, which are required for the destruction of invasive pathogens [128]. Furthermore, this activated host immune response enhances the permeability of the GIT wall, allowing these substances to pass easily into the bloodstream. This causes systematic inflammatory reactions which consequently lead to an increase in the permeability of the blood–brain barrier and activation of microglial cells [129]. Thus, bacterial invasion into the GIT causes the passage of such endotoxins as LPS or flagellin into blood circulation, leading to chronic inflammation [130,131,132,133]. Furthermore, as a physiological immune response, the blood glucose level is elevated to serve as an additional energy source for the immune cells located in the intestinal tissue [134]. In the long term, chronic inflammation in the intestinal tissue with increased blood glucose levels can lead to MetS manifesting by insulin resistance and T2D [107].

### 2.4. Gut Microbiota in Depressed Patients

The digestive system with intestinal microbiota and the brain are bidirectionally connected (as the gut–brain axis) through endocrine, immune, and neural pathways [100,135,136]. Due to this connection, the brain regulates gastrointestinal functions, e.g., peristalsis [82,100]. On the other hand, microbial imbalance (dysbiosis), influencing the gut–brain connection, may be contribute to the development of several health problems, including eating disorders, chronic abdominal pain syndrome, gastrointestinal inflammation [42,46], and various neurological diseases (i.e., Alzheimer’s disease, Parkinson’s disease, and multiple sclerosis) [137,138,139]. A high level of comorbidity between mental conditions such as depression and anxiety and gastrointestinal diseases, e.g., irritable bowel syndrome (IBS) or inflammatory bowel disease, has been observed [46]. According to the literature data [140], approximately 60% of patients with depression and anxiety complain about disturbances in gastrointestinal functions, e.g., suffer from IBS. In addition, it has been noticed that small intestinal bacterial overgrowth (SIBO) is presumably involved in the limitation of nutrient absorption in depressed patients [141]. Thus, dysbiosis in the GIT ecosystem leads to parasitic microbial overgrowth, which makes the ecosystem in general less resistant to perturbations [66]. For example, such yeast species as *C. albicans* and *Saccharomyces cerevisiae* increased their abundance in schizophrenia patients, causing dyspeptic symptoms, in comparison to non-psychiatric people [113,114]. 

The intestinal microbiota is very sensitive to both physiological and physical stress that its host experiences. Stress is a significant risk factor for depression, and it simultaneously alters the composition and proportion of intestinal microbiota, significantly decreasing levels of *Firmicutes* phylum, especially in *Bifidobacterium* and bacteria from the *Lactobacillaceae* family [46,141,142,143]. Stress causes activation of the sympathetic nervous system, and it slows down the digestive process [106,144]. Furthermore, stress can change the structure and function of the intestinal barrier, which in turn allows molecules and pathogens to easily enter the bloodstream and to induce inflammation and oxidative stress [106,140]. Apart from influencing the mucosal permeability, stress and stress mediators also have an impact on visceral sensitivity and release of neuroendocrine agents, which results in the imbalance of intestinal microbiota and weakens the host immunity, making a given person susceptible to many infections [145]. Then, stress impairs the connection between the gut microbiota and the brain, which can lead to MetS and metabolic depression [22,146,147].

Both physiological and physical stress induce stress responses related to the hypothalamic–pituitary–adrenal (HPA) axis. When a stressful situation occurs and the body detects stress mediators, the cascade of the HPA axis responses is triggered. Briefly, the hypothalamus releases the corticotrophin hormone which induces a release of the adrenocorticotrophic hormone from the pituitary glands, and as a consequence, glucocorticoids are released from the adrenal cortex. Additionally, the stress-induced secretion of pro-inflammatory cytokines takes place [106,140]. 

It is known that elevated levels of cortisol in plasma and corticotrophin-releasing factor in the cerebrospinal fluid are responsible for changes in neural activity in depressed patients. Moreover, repeated activation of these factors consequently causes an enhancement in catecholamine (mainly norepinephrine and dopamine) levels, which in turn influences several physiological functions, including attention, body posture and balance, cognitive control, emotions, and the sleep–wake cycle. Furthermore, norepinephrine has been identified as a factor that increases proliferation of pathogens such as *Escherichia coli*, *Yersinia enterocolitica*, and *Pseudomonas aeruginosa* [145]. Deficiencies in catecholamine concentration might contribute to neurodegenerative diseases such as Alzheimer’s and Parkinson’s disorders and MDD [148]. The effect of probiotics on catecholamine metabolism was determined in animal models. *Bifidobacterium infant* reversed the stress-induced decrease in norepinephrine (i.e., increased norepinephrine levels) in the brainstem of rats [149]. In turn, administration of *Lactobacillus helveticus* R0052 and *B. longum* R0175 (at a dose of 10^9^ CFU per day) reduced the stress-induced increase in plasma norepinephrine (i.e., reduced norepinephrine levels) in mice [150]. The supplementation with these probiotic strains decreased the plasma concentrations of dopamine and norepinephrine, but it did not affect these monoamines in the brain, suggesting that the probiotics would interfere with catecholamine metabolism [151]. Elsewhere, such LAB species as *Lactobacillus casei* ATG-F1, *L. reuteri* ATG-F3, and dopamine-increasing *L. reuteri* ATG-F4 strain isolated from newborn babies had potential psychobiotic properties, modulating the catecholamine pathway [152]. However, a recent clinical trial showed no significant correlations between the plasma level of catecholamine metabolites and BDNF in MDD individuals [153].

Chronic stress also reduces feedback mechanisms that normally are related to cortisol release, which in consequence causes an imbalance of the HPA axis. Activation of the HPA axis by stress exposure increases production of stress hormones, i.e., glucocorticoids and catecholamines, which increase the permeability of the intestinal barrier (“leaky gut”) [154]. Consequently, chronic stress causes intestinal dysbiosis, inflammation, and barrier dysfunction [155]. The negative alteration of the microbiota composition in the gut alters the mood and leads to physiological and psychological diseases [106,140,156]. 

Several studies have shown that the intestinal microbiota of patients with depression is significantly different from that of healthy subjects [25,106,146]. Differences in microbiota composition in depressive patients have been observed both at levels of phylum, family, and genus of bacteria. Levels of phyla *Bacteroidetes*, therein *Prevotellaceae*, and *Prevotella* and phyla *Actinobacteria* and *Proteobacteria* are usually increased, while the abundance of *Firmicutes*, *Lactobacillaceae* family, *Bifidobacterium*, *Faecalibacterium*, and *Ruminococcus* is significantly lowered in the GIT of MDD patients compared to healthy people [106,143,146]. The six biomarkers enhanced in MDD include *Proteobacteria* (*Oxalobacter* and *Pseudomonas*) and *Firmicutes* (*Parvimonas*, *Bulleidia*, *Peptostreptococcus*, and *Gemella*), while the six biomarkers in healthy controls are all derived from *Firmicutes*, including *Lachnospiraceae*, *Ruminococcaceae*, *Coprococcus*, *Blautia*, *Clostridiaceae*, and *Dorea*. This suggests that *Firmicutes* is the most significant phylum correlated to depression [59,143]. The dominant microbiota includes three major *Clostridium* clusters (IV, IX, and XIV), while other clusters have lower abundance. The significantly reduced genus from *Firmicutes* predominantly belongs to three families, which are the *Faecalibacterium* of the *Ruminococcaceae* and the *Dorea*, while the *Coprococcus* of *Lachnospiraceae* exhibit the most significant difference. These genera represent *Clostridium* clusters IV and XIVa, respectively, and can metabolize various complex carbohydrates to form different SCFAs such as acetate, butyrate, and lactate. The reduction in these fermentation-related bacteria leads to a decrease in the production of SCFAs, which in turn results in intestinal barrier dysfunction. This natural barrier function is weakened, multiple antigenic agents are exposed, and the weak GIT becomes the source of inflammation [59]. Moreover, it has been shown that the *Ruminococcaceae* family is correlated with less severe negative mental disease symptoms, while *Bacteroides* and *Coprococcus* spp. are responsible for more severe depressive symptoms. *Coprococcus* spp. is also associated with a considerable risk of coronary heart disorder [157]. Additionally, another study in MDD patients compared with healthy groups showed an increase in the relative abundance of *Eggerthella, Atopobium*, and *Bifidobacterium* and confirmed a decreased relative abundance of *Faecalibacterium* [158]. In patients suffering from depression, a high abundance of *Fusobacteria* and *Actinobacteria* at the phylum level and a high abundance of *Actinomycineae*, *Bifidobacteriaceae*, *Clostridiales* family *incertae sedis*, *Clostridiaceae*, *Eubacteriaceae*, *Fusobacteriaceae, Lactobacillaceae* XI, *Nocardiaceae*, *Porphyromonadaceae*, *Streptomycetaceae*, *Thermoanaerobacteriaceae* and low quantity of *Bacteroidaceae*, *Chitinophagaceae*, *Marniabilaceae*, *Oscillospiraceae*, *Streptococcaceae*, *Sutterellaceae*, and *Veillonellaceae* at the family level were found. Furthermore, at the genus level, a high abundance of *Actinomyces*, *Anaerofilum*, *Anaerostipes*, *Asaccharobacter*, *Atopobium*, *Blautia*, *Clostridium* IV, *Clostridium* XIX, *Desulfovibrio*, *Eggerthella*, *Erysipelotrichaceae incertae sedis*, *Eubacterium*, *Gelria*, *Holdemania*, *Klebsiella*, *Olsenella*, *Oscillibacter*, *Parabacteroides*, *Paraprevotella*, *Parasutterella*, *Parvimonas*, *Streptococcus*, *Turicibacter*, and *Veillonella* and a low abundance of *Clostridium* XlV, *Coprococcus*, *Dialister*, *Escherichia/Shigella*, *Lactobacillus*, *Howardella*, *Pyramidobacter*, and *Sutterella* were observed in depressive patients [159]. 

### 2.5. Metabolic Syndrome Microbiota

The diversity and abundance of gut microbiota are significantly different in patients with MetS associated with four sub-pathologies: obesity, dyslipidemia, increased blood sugar/insulin resistance/T2D, and increased blood pressure compared to healthy people [160]. On average, microbial metabolism participates in up to 10% of the daily calorie intake of their host; however, in obese people, this contribution is frequently higher [66,161]. This is because the abundance of bacteria responsible for conversion of non-digestible complex carbohydrates and dietary fibers into SCFAs declines in obese individuals. When the gut microbiota predominantly consists of bacteria which specialize in complex carbohydrate conversion (e.g., *Bacteroides thetaiotaomicron*) while other microorganisms rely heavily on their cohabitants to capture nutrients, providing valuable nutrients to the host is more beneficial [66,162,163]. Interestingly, transplantation of an obese gut microbiota into germ-free animals resulted in weight gain in comparison to the control group inoculated with a lean gut microbiota [164,165]. Noteworthy, that administration of SCFAs as metabiotics in the prevention and treatment of obesity induced thermogenesis in brown adipose tissue and browned white adipose tissue [166].

The *Firmicutes* to Bacteroidetes ratio was a common index to measure the structure of the gut microbiota. In research at the phylum level, an increase in the *Firmicutes* to Bacteroidetes ratio was observed in overweight/obese adults, resulting in an increased abundance of *Firmicutes* and reduced abundance of *Bacteroidetes*. However, members of *Firmicutes* show greater heterogeneity in their composition than *Bacteroidetes* in the gut of MetS patients (e.g., overweight/obesity), especially in the gut of children. For example, *Clostridium* (*Firmicutes* phylum) is positively associated with the body mass index (BMI) in children, and it is more important in young adults [61]. An increased abundance of *Clostridium*, *Dorea*, and *Ruminococcus* (*Firmicutes* phylum) was also detected in IBS [167]. Mice subjected to chronic stress had a decreased abundance of *Bacteroides* and increased abundance of *Clostridium*. In turn, rodents with a higher abundance of *Bacteroidetes* and a lower quantity of *Firmicutes* in the gut tended to manifest depressive-like behavior [156]. This was confirmed in clinical trials. *Bacteroides fragilis* is significantly associated with a higher BMI z-score in children, contributing to weight gain during childhood [61]. In the composition of the gut microbiota of obese people, some bacteria belonging to genera from the *Firmicutes* phylum, e.g., *Gemmiger*, *Coprococcus*, *Dorea*, *Faecalibacterium*, *Roseburia, and Lactobacillus*, and *Bifidobacterium* spp. (from *Actinomycetota* phylum), and from the phyla *Bacteroidetes: Alistipes* spp. as well as *Akkermansia*, and *Methanobrevibacter* (Archeae domain) exhibited significantly lower abundance than in the lean group [156,168]. These genera were highly abundant or promoted by dietary fibers as prebiotics in the lean gut microbiota, indicating their potential role in leanness [168]. Furthermore, in obese individuals, there was an increase in genera belonging to the *Firmicutes* phylum: *Ruminococcus* and *Streptococcus*; the *Bacteroidetes* phylum: *Porphyromonas*, *Bacteroides*, and *Parabacteroides*; the *Proteobacteria* phylum: *Campylobacter*; and the *Bacilliota* phylum: *Dialister* [156]. Thus, an analysis of microbiota at the phylum level/ratio is too general to notice differences in the microbiota composition in people with health problems. 

Moreover, an abundance of obesity-associated bacteria which produced branched short-chain fatty acids (BCFAs), mainly isovaleric, isobutyric, and 2-methylbutyric acids, from branched chain amino acids (leucine, isoleucine, and valine) was observed in the GIT of obese patients [169]. In the human GIT, the degradation of branched chain amino acids is mainly conducted by the genera *Bacteroides* (e.g., B. fragilis), *Clostridium*, and Propionibacterium [168,169], especially in high-protein and low-complex carbohydrate diets, such as the Western diet, resulting in higher concentrations of BCFAs in a validated in vitro gut model [168], animal study [170], and clinical trial [171]. The Western diet leads to the concomitant production of other protein degradation products such as ammonia, biogenic amines, phenol, or p-cresol which can cause cell damage in the intestinal environment [168]. The Western diet, typically consisting of high consumption of red meat, animal fat, high-sugar, and low-fiber foods, was associated with an increased quantity of *Bacteroidetes* phylum (primarily mucin-degrading bacteria) as well as *Ruminococcus* [60]. It should be emphasized that for good health, a person should consume no more than 50 g of protein per day [172], which is about 18 kg per year. However, these limits are greatly exceeded in developed countries (about 80 kg in Europe and more than 110 kg in the US and Australia per person) [173]. Furthermore, a high concentration of isovalerate and cortisol levels in feces was observed in human depressive patients [174]. In turn, isobutyrate promoted colonic Na^+^ absorption [175]. However, a recent study shows that iso-BCFAs decrease the expression of the adipocyte genes which are linked with lipid metabolism (except fatty acid synthase) and inflammation. This suggests that changes in the profile of iso-BCFAs in obese individuals may contribute to adipose tissue inflammation and dyslipidemia [176].

Obesity and depression increase the risk of development of other chronic metabolic diseases, e.g., T2D. On the one hand, depressive disorders are two times more common in patients with T2D in comparison to normoglycemic healthy people; on the other hand, depression at the baseline enhances the risk of T2D up to 60% [16]. T2D significantly increase the risk of being diagnosed with MDD, especially in women. In a cohort of clinical trials with 123,232 patients with diabetes and 1,933,218 control subjects (52% females, 48% males), women with T2D had a 2.55-fold increase in the diagnosis of MDD, with the greatest gender difference between 40 and 49 years of age compared to women without diabetes [177]. It is considered that depressive disorders are not only a direct consequence of diabetes; depression may be a risk factor for T2D [178]. Patients with T2D often show a disrespectful attitude towards their disease, with a diet deficient in prebiotic fibers, resulting in metabolic decompensation, with high and low blood glucose levels, which can cause mood alterations [17]. Furthermore, several studies have shown that intestinal dysbiosis is a factor in the rapid progress of insulin resistance in T2D, which accounts for approximately 90% of all diabetes cases worldwide. In T2D patients, a decrease in *Bifidobacterium* spp. and an increase in the abundance of *Bacteroides* spp. have been reported. Additionally, low-grade inflammation is one of the most important pathophysiological factors leading to T2D progression with hyperglycemia and insulin resistance [30]. Similarly, in IBS patients, reduction in *Bifidobacterium* spp., *Lactobacillus* spp., and *Faecalibacterium* and higher overgrowth of such pathogenic bacteria as *Campylobacter jejuni*, *Campylobacter concisus*, *C. difficile*, *Veillonella*, *Heamophilus parainfluenzae*, *Helicobacter pylori*, *Enterobacter* spp., *E. coli*, *Shigella* spp., *Salmonella* spp., and *Streptococcus* spp. were observed [179]. Noteworthy, the overgrowth of the large invasive population of *Proteobacteria* (e.g., *P. aeruginosa*, *Pseudomonas putida*, and *Klebsiella pneumoniae*) and *Bacteroidetes* (e.g., *Alistipes* spp.) reduced significantly the population of beneficial *Firmicutes* in MDD patients in comparison to the healthy group [143]. 

A diet high in fats, preservatives, and carbohydrates and low in fiber, typical of developed countries, disturbs the composition of intestinal microflora. In addition, these changes are exacerbated by stress, especially chronic stress, and the use of certain medications, including antibiotics, proton pump inhibitors, and non-steroidal anti-inflammatory drugs. Quantitative, qualitative, and functional alternations in metabolic activities of the intestinal microbiota cause the development of inflammation, which leads to such metabolic disorders as obesity or diabetes [43,76,156]. Thus, abnormal metabolic activity or a change in the composition of the gut microbiota has been proposed as a factor in higher susceptibility to disease [168], including metabolic depression. 

Dysbiosis is improved successfully by an intake of adequate amounts of oral probiotics and prebiotics, consumption of a healthy diet, and/or by transplantation of fecal microbiota [146,180]. Noteworthy, fecal microbiota transplantation from a healthy person, as an addition to treatment of psychiatric and metabolic diseases, reduced the permeability of the small intestine in NAFLD patients [181,182,183]. 

Prebiotics are compounds that induce growth or activity of the intestinal microbiota; therefore, combinations of pro- and prebiotics as synbiotics are frequently used [184,185]. For example, consumption of prebiotic inulin by obese people significantly increased butyrogenic strains, e.g., *B. adolescentis*, an unclassified *Bifidobacterium* and *Faecalibacterium prausnitzii*. Furthermore, the composition of the obese microbiota on inulin shifted the simulated intestinal environment to a healthier environment with the growth of beneficial bacteria from *Faecalibacterium*, *Blautia*, *Fusicatenibacterium*, and *Bifidobacterium* genera having the ability to regulate host health and alleviate MetS symptoms [168,186]. Moreover, it has been shown that the bifidogenic effect of inulin is more pronounced in the obese gut microbiota in comparison to lean microbiota, especially in the case of *B. animalis* and *B*. *adolescentis.* It seems that inulin can diminish the diversity within the *Bifidobacterium* genus [168]. Consequently, the health benefits of prebiotics in metabolic depression include inhibition of pathogens, immune stimulation, reduction in blood lipid levels, and insulin resistance effects. Metabolites formed after the prebiotic degradation influenced brain function, decreased blood–brain barrier permeability, and reduced neuroinflammation [184]. Furthermore, some animal studies have demonstrated that prebiotic administration reduces stress responsiveness, anxiety, and depressive-like behavior, enhances expression of BDNF, and improves cognition [187]. In clinical trials, prebiotic supplementation enhanced the levels of SCFAs [188], improved social behavior symptoms and sleep patterns in ASD [189], and caused a reduction in anxiety scores in IBS [190]. Similarly, in other clinical studies, administration of a diet high in prebiotic fibers improved mood, anxiety, stress, and sleep in adults with moderate mental stress and low prebiotic intake [191]. 

A change in the diet can affect the composition and diversity of the gut microbiota, altering the ratio of individuals genera. An increase in the population of *Clostridium* spp. and *Bilophila* spp. leads to an increase in energy efficiency, which facilitates weight gain. There are reports showing that reduced numbers of bifidobacteria in mice fed a high-fat diet increased endotoxemia. This increased endotoxemia can be reversed by prebiotic supplementation, which restores bifidobacteria levels in the mouse gut [192]. Similarly, there is evidence that the *Clostridium* and *Bilophila* genera are more prevalent in the gut of people eating a diet rich in animal-based foods, while *Bacteroides* and *Prevotella* as well as *Akkermansia* and *Bifidobacterium* are more prevalent in people eating a diet rich in plant-based foods [55]. Recently, it has also been noted that the consumption of ellagitannin-rich foods has a positive effect on both physical and mental health. Ellagitannins are found in such plants as berries, pomegranates, and nuts. Ellagitannins-rich foods may modulate signaling pathways via the intestinal microbiota-derived metabiotics. Additionally, they also contribute to the integrity of the GIT wall, influence the gut–brain axis, and as a consequence promote beneficial health effect [193].

However, the synbiotic combination of a prebiotic-rich diet and a probiotic dietary supplement (UltraBiotic 45, FiT BioCeuticals Ltd.) delivered 12 × 10^9^ CFU per capsule, containing: *Bifidobacterium bifidum* Bb-06: 2 × 10^9^ CFU; *B. animalis* subsp. *lactis* HN019: 1 × 10^9^ CFU; *B. longum* R0175: 1 × 10^9^ CFU; *L. acidophilus* La-14: 2 × 10^9^ CFU; *L. helveticus* R0052: 2 × 10^9^ CFU; *L. casei* Lc-11: 2 × 10^9^ CFU; *Lactobacillus plantarum* Lp-115: 1 × 10^9^ CFU; *Lactobacillus rhamnosus* HN001: 1 × 10^9^ CFU, which did not appear to have a beneficial effect on mental health outcomes [194]. It is worth emphasizing that the probiotic properties are strain-dependent [25,195]. It seems that the potential for mental condition improvement is strain-dependent as well. 

## 3. Probiotic Preparations in Depression

### 3.1. Probiotic Microorganisms

The World Health Organization (WHO) has defined probiotics as live microorganisms that cause health benefits in the host organism when taken in adequate amounts [196]. Fermented products were well-known and consumed by Greeks and Romans [197]. In the last couple of decades, a key role of human microbiota in both short-term and long-term health has been demonstrated unequivocally. Early microbial programming and its influence on the immune system during pregnancy, delivery, breastfeeding, and weaning are crucial, and they determine the functioning of the immune system, normal microbiome, and overall health and well-being in adult life [198,199]. Human microbiota represents a highly complex ecosystem. It has been estimated that the gastrointestinal tract in adults is composed of 100 trillion viable microorganisms [43,56,100,101,106]. Human probiotic microbiota belongs mainly to: *Bifidobacterium* and the following LAB: *Lactococcus*, genera of *Lactobacillaceae* family (now comprising of 31 genera), *Enterococcus*, and *Streptococcus* [104]. Although symbiotic and commensal microorganisms from the GIT are the main source of probiotic strains, they cannot be considered as probiotics before these strains are isolated and identified genetically and phenotypically, and their health-promoting properties are demonstrated [41,199]. 

Probiotic products support a healthy GIT by alleviating abdominal pain and bloating [2,200], flatulence, bowel irregularity and discomfort [2], IBS [100,111,201], inflammation, and oxidative stress [141], enhancing the immune system and reducing hypercholesterolemia [202]. They are also used in the prevention and treatment of the antibiotic- and *C. difficile*-associated diarrhea [111,203,204], infectious diarrhea [6,202], necrotizing enterocolitis [205], ulcerative colitis [111], and T2D [100], as well as in order to reduce an amount of enteric pathogens (including *H. pylori*) [2,141,200]. Moreover, it has been found that probiotics may exert beneficial effects in pregnancy, including prevention of gestational diabetes [206], mastitis [207], constipation [208], growth of the group B *Streptococcus* bacteria [209], and bacterial vaginosis [210]. 

The effectiveness of oral probiotics is species- and strain-dependent. Moreover, an adequate high quantity of living microorganisms, referred to as CFU, has to be administered [25,194]. Probiotics and prebiotics are considered Generally Recognized as Safe *(*GRAS**) for all age groups as well as during pregnancy and lactation [211,212]. 

### 3.2. Psychobiotics in In Vitro and In Vivo Studies

It has been proven that microbiota influences the gut–brain axis [147]. Then, administration of oral probiotics has an impact on the gut–brain communication pathways, particularly in relation to mental state and emotional behaviors [25,147]. Probiotics that improve mental conditions are also called “psychobiotics” in order to emphasize their antidepressant-like and pro-cognitive actions [2,25,147]. Supplementing an individual’s poor diet with probiotics supports treatment of depression, particularly metabolic depression [2,22]. Animal studies and clinical trials have indicated that administration of probiotics successfully reduces depressive symptoms in a similar manner to traditional antidepressant treatments [25,147]. Psychobiotics have been found primarily among specific strains of LAB and *Bifidobacterium* spp., as presented in Table 3. 

The intestinal microbiota produces several neuroactive substances, e.g., tryptophan metabolites, which can influence the brain through afferent autonomic and endocrine pathways [135]. Subspecies of the *Lactobacillaceae* family produce and secrete acetylcholine (ACh), which is known to modulate attention, learning, memory, and mood [56]. Some strains, e.g., *Bifidobacterium infantis*, *B. longum*, and *L. acidophilus*, may improve mental conditions via regulating the expression of endocannabinoids or by producing and secreting neurotransmitters such as 5-hydroxytryptamine (serotonin, 5 HT), ACh, catecholamines, dopamine, histamine, GABA, glutamate (Glu), glycine, and norepinephrine in a species-dependent manner [56,108]. These neurotransmitters modulate signaling pathways of the local enteric nervous system and then the gut–brain axis [226]. Moreover, some probiotic strains and their metabolites reverse depressive diseases via immunomodulation. Immunomodulation mechanisms predominantly include: (a) activation of macrophages by probiotic signaling, (b) stimulation of IgA-producing cells and neutrophils, (c) stimulation of peripheral Ig production, (d) stimulation of mucus production, and (e) inhibition of pro-inflammatory cytokine release inhibition [149]. For example, some strains belonging to *Lactobacillus* spp. (e.g., *L. paracasei* PS23 and *L. plantarum* 299v) reduce pro-inflammatory effects [227,228]. Additionally, some proteins such as serpins, lactocepins, and gassericens secreted by *B. longum*, *L. paracasei*, and *Lactobacillus gasseri*, respectively, act as anti-depressive factors. Serpins regulate brain activities through the vagus nerve [229]. Lactocepins suppress pro-inflammatory activities in the intestinal epithelial cells [230]. In turn, gassericins facilitate sleep via enhancement of the parasympathetic activity [231]. 

It has been observed that some *Bifidobacterium* strains enhance levels of the BDNF in the hippocampus [201]. In turn, *Lacticaseibacillus rhamnosus* (previously *Lactobacillus rhamnosus*) regulates behavioral and physical reactions via the vagus nerve, and it helps in stressful situations [56,103]. *B. infantis* 35624 consumed for 8 weeks reduced inflammatory biomarkers in patients with chronic fatigue syndrome (CFS). Furthermore, *B. bifidum*, *L. acidophilus*, *L. casei*, and *L. rhamnosus* exerted a significant effect on modulating anxiety and inflammatory processes in CFS patients [232]. However, it is difficult to determine which probiotic strain was more effective in attenuating depression symptoms because multiple strains are mostly used in clinical trials [80], and there are no studies with a placebo-controlled group evaluating the effectiveness of one probiotic strain versus another single strain in depressive disorders. However, it is possible to determine the properties of psychobiotics that they should possess, as shown in Figure 1.

### 3.3. Meta-Analysis Results on Probiotics in Depression

The impact of probiotics supplementation on depressed mood was assessed in several meta-analyses. Meta-analyses evaluating data obtained between 2016 and 2023 showed that probiotics reduced the depression scale score by 0.24–1.62 when given as monotherapy or as an add-on to standard antidepressant therapy (TAU) [2,35,36,233,234,235,236,237]. In turn, other meta-analyses indicated that probiotics did not affect anxiety symptoms and mood both in diagnosed patients and in healthy subjects [35,56,235,238].

Recently, Ng et al. [235] have reviewed 10 double-blind, randomized, placebo-controlled trials (RTCs) with a total of 1349 patients. In this meta-analysis, the authors did not reveal any positive effect of probiotic administration on the mood of both healthy and depressive populations. However, it was observed that the depression scale score was reduced by 0.684 in mild and moderate depression [235]. In turn, Reis et al. [238], who analyzed 22 preclinical studies (743 animals) and 14 clinical trials (1527 participants), and Chao et al. [35], who analyzed 10 RCTs with 656 subjects, indicated that the use of the probiotics did not exert an impact on anxiety symptoms. Moreover, not all researchers supported the theory of the positive impact of probiotic supplementation in patients suffering from depression, e.g., Romijn et al. [214], who administrated *L. helveticus* R0052 and *B. longum* R0175 as monotherapy in 79 patients with moderate depression, pointed out that there were no significant differences between the probiotic and placebo groups on any psychological outcome measure. Additionally, a recent clinical trial showed that no positive or negative emotional stimuli were observed after supplementation of a probiotic dietary supplement (Bio-Kult Advanced), consisting of 14 bacterial species (Table 3), encapsulated at 2 × 10^9^ CFU/capsules per 4 weeks, with conventional antidepressant therapy (TAU). Nevertheless, the authors of this trial emphasized that the administration of a probiotic may be regarded as an “early intervention” strategy to reduce the risk of developing MDD in individuals with mild to moderate depression [225]. 

In contrast, in a meta-analysis performed by Goh and co-workers [233], who have evaluated 19 RTCs with 1901 participants, it was noted that patients receiving probiotic supplementation presented a greater reduction in symptoms of MDD as compared to placebo-treated individuals. Notwithstanding, both of Ng and co-workers’ [235] and Goh and co-workers’ meta-analyses [233] indicated inconsiderable mood amelioration in subjects with pre-existing depressed mood treated with probiotics, whereas such changes were not noted in healthy participants or in patients with other clinical conditions. Moreover, Goh and co-workers [233] found that administration of single-strain probiotics is less effective in reducing depressive manifestations than administration of multiple-strain probiotics. The most comprehensive meta-analysis that showed general beneficial effects of probiotics supplementation in depression was published by Liu and colleagues [234]. The authors indicated that: (1) *Lactobacillus* was the most popular among the tested probiotics, (2) *Lactobacillus* had no effect on depression when used alone, and (3) there were significant differences between research where *Lactobacillus* was used as monotherapy versus those in which it was concomitantly supplemented with other probiotics and/or prebiotics [234]. Several recent systematic reviews on outcomes from clinical trials evaluating the influence of probiotics on low mood also supported their use in patients with depression. Their main findings are the following: (1) probiotics have a beneficial effect on depression behaviors, and (2) *L. helveticus*, *L. rhamnosus*, and *L. casei*, as well as *B. longum*, *B. breve*, and *B. infantis*, are the most effective in improving depressive symptoms [110,239]. Similar conclusions in reducing the depression scale scores were also drawn in other meta-analyses from the last five years [35,36,236,240,241]. 

Furthermore, Nikolova and her co-workers [236] conducted a meta-analysis of seven RTCs with 404 subjects and found that probiotics reduced the depression scores by 0.83 effectively only when administered as an additive to TAU compared to the placebo + TAU treatment group. There was no significant probiotic impact as a standalone treatment. These results are confirmed in the most recent analysis carried out by Forth and co-workers [242]. The authors analyzed five RTCs with MDD and one Generalized Anxiety Disorder (GAD). The study demonstrated that the adjuvant probiotic or synbiotic treatment was more efficacious in improving the symptoms of psychiatric disorders than the first-line treatment (TAU) alone or with placebo. The antidepressant with probiotic adjuvant treatment may be beneficial for improving antidepressant tolerability. 

Interestingly, in the recent meta-analysis conducted by Sikorska and co-workers [243], based on 19 RTCs with 1406 participants, the positive effect of specific probiotics was noticed only in a group of patients suffering from depression. The probiotics supplemented to healthy people did not exert an effect. Moreover, in an umbrella meta-analysis (*n* = 10) with 8886 participants reported by Musazadeh and co-workers [80], it was demonstrated that probiotics administered in an appropriate dose for a sufficiently long time had a significant effect on reducing depression. The specific probiotics should be administered at a dose of >10^10^ CFU per day for >8 weeks to alleviate the symptoms of depression. Therefore, additional treatment with probiotics ameliorates depressive symptoms along with changes in gut and brain microbiota, highlighting the role of the microbiota–gut–brain axis in MDD and emphasizing the potential of microbiota-related treatments as accessible, pragmatic, and non-stigmatizing therapies in MDD [223]. Furthermore, in the meta-analysis conducted by Zhang and co-workers [118] based on 25 RTCs with 1,931 adult people over 18 years of age, it was detected that the consumption of probiotics significantly reduced body weight and body mass index (BMI). The best effect was achieved in the population of overweight and obese people, as the supplementation of the probiotics lasted longer than 8 weeks. This meta-analysis also suggests a better impact of multi-strain probiotics in the reduction in body weight and BMI. Supplementation with specific probiotics has a beneficial effect on both anthropometric and metabolic parameters [76].

It can be assumed that not all probiotic strains have an impact on mental health. Hence, probiotics with psychobiotic properties should be evaluated in clinical trials. Thus, taking into consideration the beneficial effect of probiotics on depressive symptoms, specific psychobiotics might be administered in a daily dose not less than 10 billion CFU and for a sufficiently long period exceeding 8 weeks as an additive to antidepressants in routine clinical treatment of mental diseases, especially in metabolic depression.

### 3.4. Probiotics in Metabolic Depression

There are insufficient numbers of studies on the effect of probiotics on metabolic depression. In animal models, *L. rhamnosus* JB-1 prevented some antibiotic-induced changes in rodents, e.g., depression-like behavior [244]. In another preclinical study, *L. fermentum* MCC2759 and MCC2760 strains alleviated inflammation, improved intestinal barrier integrity, and improved insulin sensitivity in diet- and streptozotocin-induced rat diabetes with metabolic abnormalities [245]. Elsewhere, *L. fermentum* CECT5716 exerted anti-obesity effects, in relation to its anti-inflammatory function, and ameliorated endothelial dysfunction and intestinal dysbiosis in a study of high-fat diet-induced obesity in mice [246]. Furthermore, administration of GABA-producing *L. brevis* DPC6108 and *L. brevis* DSM32386 improved both metabolic abnormalities and depressive-like behavior associated with MetS in mice [247].

The effect of the use of probiotics on the metabolic depression subpopulation in clinical trials is presented in Table 4.

Administration of *B. longum* NCC3001 to depressive patients with IBS reduced responses to negative emotional stimuli in multiple brain areas, including the amygdala and fronto-limbic regions, in comparison to placebo controls. Both the probiotic and placebo groups had similar profiles of fecal microbiota, serum markers of inflammation, and concentration of neurotrophins and neurotransmitters, but patients receiving *B. longum* NCC3001 had reduced urine levels of methylamines and aromatic amino acid metabolites. Patients took the probiotic/placebo for 6 weeks. The levels of anxiety, depression, IBS symptoms, and quality of life were determined after 10 weeks. It was shown that the reduced metabolic depression scores improved the quality of life of patients taking the probiotic versus the placebo. It was detected that these improvements were associated with changes in brain activation patterns indicating that *B. longum* NCC3001 reduced limbic reactivity [201]. In another clinical trial with MDD patients with IBS, supplementation of *B. coagulans* MTCC 5856 significantly reduced serum myeloperoxidase, i.e., an inflammatory biomarker compared with the baseline and placebo controls. *B. coagulans* MTCC 5856 was found to be efficient in the improvement of depression and IBS symptoms [248]. 

Only one clinical trial recruited pregnant women with gestational diabetes mellitus (GDM) having problems with mental health and life quality. It showed that supplementation of a mixture of four strains: *L. acidophilus* LA-5, *Bifidobacterium* BB12, *Str. thermophilus* STY-31, and *L. delbrueckii* subsp. *bulgaricus* LBY-27, improved the quality of life and reduced depression levels in the pregnant women with GDM [249]. Furthermore, supplementation of selected multiple psychobiotic preparation (Biocult strong) containing seven different strains of LAB and *Bifidobacterium lactis CNCM I-2494* (Table 4) in combination with hypocaloric diet or prebiotics seems to solve the problems related to obesity and behavior disorders [250,251]. Interestingly, in another clinical trial, 12-week co-supplementation of a probiotic containing three different strains (Table 4) and selenium to Alzheimer’s disease patients significantly improved their cognitive function and some metabolic profiles [252].

Further studies are needed to demonstrate the effectiveness of probiotics in metabolic depression. Currently, research mainly focuses on *Bifidobacterium* and *Lactobacillus* strains against depression and MetS. However, the next-generation microorganisms [22], e.g., *Akkermansia muciniphila* and *Faecalibacterium prausnitzii*, as well as some beneficial strains of *Bacillus, Blautia,* and/or *Fusicatenibacterium*, are waiting to be evaluated for their effectiveness and application in therapy.

## 4. Conclusions

Evaluation of the microbiota–gut–brain axis, as well as assessment of a relationship between the intestinal microbiota and psychiatric disorders, including depression, has become more and more popular among scientists in recent years. In addition, an administration of substances/factors such as probiotics and their metabolites that may improve functioning of the gut–brain connection is considered as a new relevant approach to the management of mental diseases, especially metabolic depression. It is known that supplementation of probiotics (with or without the addition of prebiotics and healthy diet) has a well-established status in the treatment of gastrointestinal dysbiosis. Furthermore, probiotics with psychobiotic properties used as an adjuvant to antidepressant treatment seem to play a significant role in the management and therapy of mental and emotional health, especially in MDD and metabolic depression. Moreover, psychobiotics and metabiotics used as adjunctive preparation to antidepressant therapy combined with a high-fiber diet, ellagitannin-rich foods, and moderate-intensity exercise, and some trace elements as selenium and/or such vitamins as B_12_ and D, seem to be more helpful and efficient in metabolic depression. In the case of mild depression, psychobiotic monotreatment with psychological, natural, and alternative therapy may be sufficient to alleviate depression symptoms and may act as an important role in the prevention of depression. It seems that psychobiotic properties which relieve depression symptoms are strain-dependent. Most of all, psychobiotics must be used in the proper dose of not less than 1 billion CFU, preferably above 10 billion CFU per day, for a sufficiently long time (at least 8 weeks) in order to be effective. 

However, there is a major limitation of the analysis of psychobiotic properties. Most studies are mainly based on different species of the genera *Bifidobacterium* and *Lactobacillus*. More research is needed on other beneficial gut bacterium species of the genera such as *Akkermansia*, *Bacillus*, *Blautia*, *Faecalibacterium*, and *Fusicatenibacterium*.

## Figures and Tables

**Figure 1 molecules-28-03213-f001:**
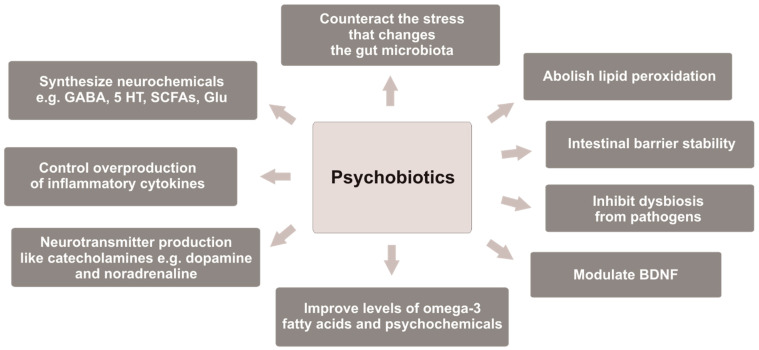
Properties of psychobiotics that may be responsible for alleviation of depression symptoms. GABA—gamma-aminobutyric acid; 5 HT—5-hydroxytryptamine; SCFAs—short-chain fatty acids; Glu—glutamate; BDNF—brain-derived neurotrophic factor.

**Table 1 molecules-28-03213-t001:** Neuroactive metabiotics of intestinal microbiota in a healthy organism and their effects on depression and metabolic depression.

Metabiotics		Gut Microbiota	Metabiotic Effect on the Host Organism	Ref.
Short-chain fatty acids (SCFAs)	Acetate	*Akkermansia muciniphila* *Bacteroides* spp. *Bifidobacterium* spp. *Prevotella* spp. *Ruminococcus* spp.	Inhibition of the production of pro-inflammatory cytokines and chemokines by AhR agonists. Maintenance of the microglial homeostasis by AhR (in vitro). Stimulation of the secretion of intestinal PYY hormones and GLP-1 resulting in decreased appetite. Enhancement of leptin production. Precursors for cholesterol and fatty acids synthesis.	[69,75,76]
	Butyrate	*Anaerostipes* spp. *Bifidobacterium infantis* *Butyricicoccus pullicaecorum* *Coprococcus catus* *Coprococcus comes* *Coprococcus eutactus* *Clostridium tyrobutyricum* *Eubacterium hallii* *Eubacterium rectale* *Faecalibacterium prausnitzii* *Lactobacillus paracasei* *Lactobacillus plantarum* *Roseburia* spp.	Inhibition of the production of pro-inflammatory cytokines and chemokines by AhR agonists. Suppression of the HDAC activity. Reduction in intestinal permeability and inflammation. Microglial homeostasis maintenance by AhR. Inhibition of the lysolecithin-induced demyelination and enhancement of remyelination (in vitro). Stimulation of the secretion of intestinal PYY hormones and GLP-1 resulting in decreased appetite. Inhibition of fat accumulation in adipocytes. Enhancement of leptin production. Substrate of gluconeogenesis.	[47,69,75,76,77,78,79,80,81]
	Propionate	*Bacteroides* spp. *Phascolarctobacterium succinatutens* *Dialister* spp. *Veillonella* spp.	Inhibition of the production of pro-inflammatory cytokines and chemokines by AhR agonists. Suppression of the HDAC activity. Regulation of microglial homeostasis. Differentiation of Treg-cell. Reduction in the production of IL-12. Enhancement of IL-10 production. Stimulation of fat storage in adipose tissue. Stimulation of intestinal epithelial integrity. Enhancement of the oxidation of fatty acids. Stimulation of mucin production.	[69,75,76,77]
	Lactate	*Bacteroides* spp. *Bifidobacterium adolescentis* *Lactobacillus* spp.	Promotion of brain health during exercise. Induction of the expression of immediate early genes and cerebral angiogenesis. Substrate for conversion into butyrate and propionate.	[82,83]
Tryptophan metabolites	Indole-3-acetic acid	*Bacteroides* spp. *Bifidobacterium adolescentis* *Bifidobacterium. longum* *Bifidobacterium pseudolongum* *Clostridium* spp. *Enterobacter cloacae* *Lactobacillus* spp.	Reduction in the production of pro-inflammatory cytokines by AhR ligands. Attenuation of the severity of intestinal inflammation.	[70,84,85]
	Indole-3-aldehyde	*Lactobacillus acidophilus* *Lactobacillus reuteri*	AhR ligands. Maintenance of intestinal homeostasis by an increase in AhR-dependent interleukin-22 transcription. Activation of cell lymphoids and gaining resistance against pathogens.	[41,84,86]
	Indole-3-propionic acid	*Clostridium* spp. *Peptostreptococcus* spp.	AhR ligands and a free radical scavenger. Protection against amyloid β in Alzheimer’s disorder. Help in better insulin secretion and sensitivity and reduction in type 2 diabetes.	[84,87,88]
	Indole acrylic acid	*Clostridium sporogenes* *Peptostreptococcus* spp.	AhR ligands. Anti-inflammatory function and enhancement of the intestinal epithelial barrier.	[84,89]
Lipoteichoic acid		*Bifidobacterium animalis*	Fat-reducing properties by fat deposition via the IGF-1 pathway.	[90]

AhR—aryl hydrocarbon receptor; HDACs—histone deacetylases; IGF-1—insulin-like growth factor-1; IL—interleukin; PYY—peptide YY.

**Table 2 molecules-28-03213-t002:** Probiotic properties that are beneficial to health.

Probiotic Properties	Ref.
Antagonistic activity towards enteric pathogens Production of antimicrobial substances	[104,105]
Competition with pathogens for adhesion to the epithelium	[104,105]
Impact on development, maturation, and modulation of the immune system (augmentation of the nonspecific and antigen-specific defenses against infections and tumors, increased production of immunoglobulins, enhanced activity of macrophages and lymphocytes), immunomodulation	[100,104,105]
Inhibition of bacterial toxin production	[104]
Intestinal synthesis and metabolism of certain neurotransmitters: aminobutyric acid, serotonin, dopamine, noradrenaline, melatonin, histamine, and acetylcholine	[106,107,108,109]
Impact on the intestinal wall integrity and enhancement of the intestinal mucosal barrier	[100,105,110]
Maintenance of normal levels of short-chain fatty acids (SCFAs)	[111]
Stimulation of the regeneration of intestinal epithelial cells	[110]
Antihypertensive effect	[110]
Anticancer effect towards colon cancer and antimutagenic activity	[110]

**Table 3 molecules-28-03213-t003:** Summary of probiotic strains used as psychobiotics in clinical trials on depression (last five years).

Probiotic Strains + Other Active Ingredients	CFU/g	Trial Designer/Clinical Outcome(s)	Daily Dose/Duration/Intervention Type/Sample Groups	Effectiveness	Ref.
*Lactobacillus acidophilus* *Lactobacillus casei* *Bifidobacterium bifidum* *Lactobacillus fermentum*	2 × 10^9^	RTC/Depression, anxiety, and stress in multiple sclerosis patients	One capsule/12 weeks/monotherapy/probiotic (*n* = 30) or placebo (*n* = 30)	Reduction in depression scale score, inflammatory factors, markers of insulin resistance, HDL-, total-/HDL-cholesterol	[213]
*Lactobacillus rhamnosus* HN001	6 × 10^9^	RTC/Pregnancy and post-partum symptoms of maternal depression and anxiety in the post-partum period	One capsule/45 weeks/monotherapy/probiotic (*n* = 193) or placebo (*n* = 187)	Reduction in depression and anxiety scores;	[39]
*Lactobacillus helveticus* R0052 *Bifidobacterium longum* R0175	≥2 × 10^9^	RTC/Moderate depression	1.5 g sachet/8 weeks/monotherapy/probiotic (*n* = 40) or placebo (*n* = 39)	No significant difference between probiotic and placebo groups on any psychological outcome measure or any blood-based biomarker	[214]
*Bifidobacterium breve* *Bifidobacterium longum* *Lactobacillus acidophilus* *Lactobacillus bulgaricus* *Lactobacillus casei* *Lactobacillus rhamnosus* *Streptococcus thermophilus* 100 mg fructooligosaccharide	2 × 10^8^ 1 × 10^9^ 2 × 10^8^ 2 × 10^9^ 3 × 10^8^ 3 × 10^8^ 3 × 10^8^	RTC/Moderate depression	500 mg/6 weeks/add on Fluoxetine/synbiotic (*n* = 20) or placebo (*n* = 20)	Reduction in HAM-D score	[215]
*Lactobacillus helveticus* *Bifidobacterium longum*	≥2 × 10^9^	RTC/MDD	5 g sachet/8 weeks/add on/probiotic (*n* = 28) or prebiotic (galactooligosaccharide) (*n* = 27) or placebo (*n* = 26)	Decrease in BDI score and kynurenine/tryptophan ratio, increasing the tryptophan/isoleucine ratio increased in in only probiotic group; no significant effect of prebiotic supplementation	[216]
*Bifidobacterium breve* A-1	5 × 10^10^ (10^11^/day)	Open-label, single-arm study/anxiety and depressive symptoms in patients with schizophrenia	Two sachets per 2 g/4 weeks/monotherapy/probiotic (*n* = 24)	Improvement of HADS PANSS score; reduction in anxiety and depressive symptoms in patients with schizophrenia; reduction in intake of dairy products	[40]
“OMNi-BiOTiC^®^ Stress Repair” (Winclove BV): *Bifidobacterium bifidum* W23 *Bifidobacterium lactis* W51 *Bifidobacterium lactis* W52 *Lactobacillus acidophilus* W22 *Lactobacillus casei* W56 *Lactobacillus paracasei* W20 *Lactobacillus plantarum* W62 *Lactobacillus salivarius* W24 *Lactobacillus lactis* W19 125 mg D-Biotin (vitamin B7) 30 mg of common horsetail 30 mg of fish collagen 30 mg of keratin plus matrix	7.5 × 10^9^	RTC/MDD	3 g/28 days/monotherapy/preparate (*n* = 42) or placebo (*n* = 40)	Improvement in BDI scores; increase in inflammation-regulatory and metabolic pathways; increase in microbial diversity profile of gut microbiota and increase in the abundance of *Ruminococcus gauvreauii* and *Coprococcus*	[217]
“CEREBIOME” (Lallemand Health Solutions Inc.): *Lactobacillus helveticus* R0052 (Rossel-52) *Bifidobacterium longum* R0175 (Rossel-175)	3 × 10^9^	Open label clinical trial/MDD	One sachet per 1.5 g/8 weeks/monotherapy/probiotic (*n* = 10)	Reduction in clinical symptoms of depression; improvement of sleep quality	[218]
*Clostridium butyricum* MIYAIRI 588 (CBM588)	No data	Open label clinical trial/treatment-resistant MDD	60 mg per day/8 weeks/add on combination with antidepressants (flvoxamine, paroxetine, escitalopram, duroxetine, and sertraline)/probiotic (*n* = 20) or placebo (*n* = 20)	Improvement of BDI and BAI scale scores; reduction in depression symptoms	[219]
*Lactobacillus plantarum* PS128	3 × 10^10^	RTC/ASD in boys 7–15 ages	28 days/monotherapy/probiotic (*n* = 36) or placebo (*n* = 35)	Amelioration of opposition/defiance behaviors	[220]
“Ecologic Barrier” (Winclove BV): *Bifidobacterium* bifidum W23 *Bifidobacterium lactis* W51 *Bifidobacterium lactis* W52 *Lactobacillus acidophilus* W37 *Lactobacillus brevis* W63 *Lactobacillus casei* W56 *Lactobacillus salivarius* W24 *Lactococcus lactis* W19 *Lactococcus* lactis W58	2.5 × 10^9^ (10^10^/day)	RTC/mild to severe depression MDD	Two sachets per 2 g/8 weeks/monotherapy/probiotic (*n* = 34) or placebo (*n* = 37)	Reduction in cognitive reactivity, improvement in BDI; no significant alteration of the microbiota in depressed individuals	[221]
*Lactobacillus reuteri* NK33 *Bifidobacterium adolescentis* NK98	2.0 × 10^9^ 0.5 × 10^9^ (2.5 × 10^9^/day)	RTC/mental health, sleep quality in healthy adults	One capsule per 500mg/8 weeks/monotherapy/probiotic (*n* = 78) or placebo (*n* = 78)	Reduction in depressive symptoms at 4 and 8 weeks of treatment and anxiety symptoms at 4 weeks; improvement in sleep quality, decrease in serum interleukin-6 levels; increase in the abundance of *Bifidobacteriaceae* and *Lactobacillacea* but decreasing *Enterobacteriaceae* in the gut microbiota composition	[222]
“Vivomixx” (Mendes SA): *Streptococcus thermophilus* NCIMB 30438, *Bifidobacterium breve* NCIMB 30441, *Bifidobacterium longum* NCIMB 30435 (re-classified as *B. lactis*), *Bifidobacterium infantis* NCIMB 30436 (re-classified as *B. lactis*), *Lactobacillus acidophilus* NCIMB 30442 *Lactobacillus plantarum* NCIMB 30437 *Lactobacillus paracasei* NCIMB 30439 *Lactobacillus delbrueckii* subsp. *bulgaricus* NCIMB 30440 (re-classified as *L. helveticus*).	(9 × 10^11^/day)	RTC/MDD	31 days/add on TAU/probiotic (*n* = 21) or placebo (*n* = 26)	Amelioration of depressive symptoms (HAM-D); decline in the BDI score; no significant effect on sleep; reduction in *Enterobacteriaceae*, *Muribaculaceae*, *Peptostreptococcaceae*, and *Veilonellaceae* populations in the intestine; reduction in the ratios of *Enterobacteriaceae* to *Bifidobacteriaceae* and *Enterobacteriaceae* to *Lactobacillaceae*	[223]
“Vivomixx” another brand name “Visbiome” (Mendes SA): *Streptococcus thermophilus* NCIMB 30438, *Bifidobacterium breve* NCIMB 30441, *Bifidobacterium longum* NCIMB 30435 (re-classified as *B. lactis*), *Bifidobacterium infantis* NCIMB 30436 (re-classified as *B. lactis*), *Lactobacillus acidophilus* NCIMB 30442 *Lactobacillus plantarum* NCIMB 30437 *Lactobacillus paracasei* NCIMB 30439 *Lactobacillus delbrueckii* subsp. *bulgaricus* NCIMB 30440 (re-classified as *L. helveticus*).	(9 × 10^11^/day)	RTC/cognitive symptoms in MDD	4 weeks (31 days)/add on TAU/probiotic (*n* = 30) or placebo (*n* = 30)	Improvement in Verbal Learning Memory Test	[224]
“Bio-Kult” Advanced (ADM Protexin Ltd.) *Bacillus subtilis* PXN 21 *Bifidobacterium bifidum* PXN 23 *Bifidobacterium breve* PXN 25 *Bifidobacterium infantis* PXN 27 *Bifidobacterium longum* PXN 30 *Lactobacillus acidophilus* PXN 35 *Lactobacillus delbrueckii* ssp. *bulgaricus* PXN 39 *Lactobacillus casei* PXN 37 *Lactobacillus plantarum* PXN 47 *Lactobacillus rhamnosus* PXN 54 *Lactobacillus helveticus* PXN 45 *Lactobacillus salivarius* PXN 57 *Lactococcus lactis* ssp. *lactis* PXN 63 *Streptococcus thermophilus* PXN 66	2 × 10^9^ CFU/capsule	RTC/emotional salience and mood in subjects with moderate depression	4 weeks/add on TAU/probiotic (*n* = 51) or placebo (*n* = 51)	Increase in accuracy at identifying facial expressions, reduction in reward learning; other aspects of cognitive performance were not affected; salivary cortisol or circulating CRP concentrations were not altered; reduction in depression scores on the Patient Health Questionnaire-9	[225]

ASD—autism spectrum disorder; BDI—Beck Depression Inventory; BAI—Beck Anxiety Inventory; CFU—colony forming unit; CRP—C-reactive peptide; HADS—Hospital Anxiety and Depression Scale; HAM-D—Hamilton Depression Rating Scale; IBS—irritable bowel syndrome; MADRS—Montgomery–Åsberg Depression Rating Scale; MDD—major depressive disorder; PANSS—Positive and Negative Syndrome Scale; RTC—double-blind, randomized, placebo-controlled trial; TAU—treatment as usual.

**Table 4 molecules-28-03213-t004:** Summary of the effectiveness of probiotic strains used as psychobiotics in clinical trials on metabolic depression.

Probiotic Strains + Other Active Ingredients	CFU/g	Trial Designer/Clinical Outcome(s)	Daily Dose/Duration/Intervention Type/Sample Groups	Effectiveness	Ref.
*Bifidobacterium longum* NCC3001 (BL)	1 × 10^10^	RTC/Mild and moderate anxiety and depression in patients with IBS and diarrhea or a mixed-stool pattern	1 g powder/6 weeks/monotherapy/probiotic (*n* = 22) or placebo (*n* = 22)	Reduction in depression; no significant impact of anxiety scores and IBS symptoms; increasing quality of life in patients with IBS.	[201]
*Bacillus coagulans* MTCC 5856	2 × 10^9^ spores = 333.33 mg	RTC/MDD in IBS patients	600 mg tablets/90 days/monotherapy probiotic (*n* = 20) or placebo (*n* = 20)	Reduction in the depressive and IBS scale score, increasing quality of life in patients with IBS.	[248]
“4biocap” (Kristin Hansen): *Lactobacillus acidophilus* LA-5 *Bifidobacterium* BB12 *Streptococcus thermophilus* STY-31 *Lactobacillus delbrueckii* subsp. *bulgaricus* LBY-27	>10^9^	RTC/quality of life depression in pregnant women with gestational diabetes mellitus	One capsule per 180 mg/8 weeks/monotherapy/probiotic (*n* = 32) or placebo (*n* = 32)	Reduction in the depressive scale score; improvement of the physical dimension of quality of life; increase in mean of total quality of life.	[249]
“Biocult strong” (HOMEOSYN) *Bifidobacterium lactis CNCM I-2494* *Lactobacillus acidophilus* *Lactobacillus bulgaricus CNCM I-1632* *Lactobacillus bulgaricus CNCM I-1519* *Lactobacillus lactis subspecies lactis CNCM I-1631* *Lactiplantibacillus plantarum* *Limosilactobacillus reuteri DSM 17938* *Streptococcus thermophilus CNCMI-1630*	1.5 × 10^10^	RTC/Anxiety and obesity	One bag of 3 g/3 weeks/psychobiotic (*n* = 15) or psychobiotic with a hypocaloric diet as synbiotic (*n* = 15) or only hypocaloric diet (*n* = 15)	Reduction in HAM-A only in psychobiotic and synbiotic groups; reduction in body composition parameters: weight, BMI, waist circumference, TBFat and IMAT in hypocaloric diet and synbiotic groups; reduction in hip circumference and TBLean only in hypocaloric diet group; reduction in II in psychobiotic and synbiotic groups; reduction in waist/hip ratio, ABFat, and GBFat only in synbiotic group.	[250]
*Lactobacillus rhamnosus* CGMCC 1.3724 (LPR) 210 mg of oligofructose 90 mg of inulin	1.6 × 10^8^	RTC/Appetite sensations and eating behaviors in obese patients in the context of a weight-reducing program	Two capsules/24 weeks/add on a personalized diet plan targeting a 2092 kJ/day (500 kcal/day) energy restriction/synbiotic obese men (*n* = 45) and women (*n* = 60)	Decrease in the BDI score; increasing weight loss only in women; benefit effect on fasting fullness and cognitive restraint.	[251]
“Bifiform Balance” (Ferrosan A/S, Pfizer): *Lactobacillus rhamnosus GG* *Bifidobacterium animalis* subsp. lactis Bb12	10^9^ 10^9^	Normalization of *Candida albicans* in male schizophrenia patients	One capsule/14 weeks/monotherapy/male probiotic (*n* = 22) or placebo (*n* = 15)	Reduction in *C. albicans* antibodies, ameliorating *C. albicans*-associated gut discomfort; no significant change PANSS scores.	[102]
*Lactobacillus acidophilus*, *Bifidobacterium bifidum*, *Bifidobacterium longum* selenium (200 μg/day)	2 × 10^9^ 2 × 10^9^ 2 × 10^9^	RTC/Alzheimer’s disease by correcting metabolic abnormalities and attenuating inflammation and oxidative stress	12 weeks/monotherapy/probiotic with Se (*n* = 26) or placebo (*n* = 26)	Increase in mini-mental state examination score; reduction in serum high sensitivity C-reactive protein, homeostasis model of assessment-insulin resistance, LDL-cholesterol, and total-/HDL-cholesterol ratio, increasing total glutathione, quantitative insulin sensitivity check index, and total antioxidant capacity; reduction in insulin levels.	[252]
“Familact” (Zist Takhmir Pharmaceutical Co.) *Lactobacillus casei* *Lactobacillus acidophilus* *Lactobacillus rhamnosus* *Lactobacillus bulgaricus* *Bifidobacterium breve* *Bifidobacterium longum*, *Streptococcus thermophilus* fructo-oligosaccharides (FOS)	10^9^	RTC/hypothyroid patients with depression	500 mg/10 weeks/monotherapy/synbiotic (probiotics + fructo-oligosaccharide, *n* = 28) or placebo (*n* = 28)	No favorable effect on depression and TSH, but it improved blood pressure and quality of life in patients with hypothyroidism.	[253]

ABFat—android body fat; BDI—Beck Depression Inventory; HAM-A—Hamilton anxiety rating scale; II—impedance index; GBFat—gynoid body fat; IMAT—intermuscular adipose tissue; PANSS—Positive and Negative Syndrome Scale; RTC—double-blind, randomized, placebo-controlled trials; TBFat—total body fat; TBLean—total body lean; TSH—thyroid-stimulating hormone.

## Data Availability

Not applicable.

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
