# Peer review of "The Role of Probiotics and Their Metabolites in the Treatment of Depression"

_molecules, 2023, doi:10.3390/molecules28073213_

Round 1
Reviewer 1 Report
This paper reviews the role of probiotics and their metabolites in the treatment of depression as recent studies revealed a relationship between gut microbiota and brain via the gut-brain axis.
The topic is important and the manuscript provides a broad analysis of the subject. However, it seems to me that there is too much information with no clear structure, therefore, the ideas are difficult to understand.
At this stage, the manuscript has no unity, it looks like the different sections were not analyzed, discussed, and appraised by all the authors.
I have the following suggestions:
- the whole manuscript should be restructured
- sections 2 and 3 can be merged; some data repeat in the two sections, some info is redundant
- in vivo and clinical experiments should be presented and discussed separate
- why is “3.1. Mechanism of probiotics action in the metabolic syndrome” here? It should rather be “mechanisms of microbiota in gut-brain axis”. Just an example of such “ellagitannin-rich foods, through gut microbiota-derived metabolites, could modulate signaling pathways, contribute to the intestinal wall integrity, act on the gut–brain axis, and promote beneficial health effects (doi: 10.3390/foods12020270)”
- the section “4.1. Treatment of depression” can be deleted
Author Response
List of responses to each of the Reviewer #1 comments
We appreciate Reviewer’s suggestions. They are unbelievably valuable for our manuscript.
According to recommendations of Reviewer #1 we revised our manuscript to meet all requirements:
Reviewer’s comments:
- The whole manuscript should be restructured
We restructured the whole manuscript according to Reviewer’s requirements stated below.
- Section 2 and 3 can be merged; some date repeat in the two sections, some info is redundant
Sections 2 and 3 were merged, and we removed repeating information from these sections.
- In vivo and clinical experiment should be presented and discussed separate.
We separated information on animal studies from human studies and discussed them separately. Now, it is clear which information relates to non-clinical studies and which information relates to clinical studies.
- Why is “3.1. Mechanism of probiotics action in the metabolic syndrome here? I should be rather be “mechanisms of microbiota in gut-brain axes”. Just an example of such “ellagitannin-rich foods, through gut microbiota-derived metabolites, could modulate signaling pathways, contribute to the intestinal wall integrity, act on the gut-brain axis, and promote beneficial health effects (doi: 10.3390/foods12020270)”
We changed the name of the subheading from “Mechanism of probiotics action in the metabolic syndrome” to “Mechanisms of microbiota in gut-brain axes”. Now, it is a section 2.3. We added a new section 2.5, in which we discussed the article on ellagitannin-rich foods (doi: 10.3390/foods12020270) in lines 575-580. It is reference No 193.
- the section “4.1. Treatment of depression” can be deleted
- We removed the section 4.1.
Reviewer 2 Report
Excellent and comprehensive Review. Minor spelling check out should be performed.
Author Response
List of responses to each of the Reviewer #2 comments
We appreciate Reviewer’s suggestions. They are very valuable for our manuscript.
According to recommendations of Reviewer #2 we revised our manuscript to meet all requirements:
Excellent and comprehensive Review. Minor spelling check out should be performed.
The native speaker checked English language and style.
Reviewer 3 Report
It was a great pleasure for me to read a brilliant and very carefully prepared review on the role of probiotics in the treatment of depression. In fact, the review takes up a much deeper topic of the relationship between gut and brain function. And this axis is not only limited to mental pathology. It is also realized through impaired liver function and various variants of neuro- and endocrine pathology. This has become especially relevant after the COVID-19 pandemic. It is known that damage to the microbiota during coronavirus infection occurs directly and indirectly, and intestinal dysbiosis and leaky gut syndrome are the key causes of post-COVID syndrome. This rich topic deserves a separate review, and it would be great if the professional team of the authors would consider writing such an article. This review contains a large number of references, is excellently structured, easy to read and contains a wealth of information useful to both scientists and medical practitioners. For ease of perception, the material is summarized in several tables and one diagram. The article may be published in present form.
Author Response
List of responses to each of the Reviewer #3 comments
We appreciate Reviewer’s suggestions. They are very valuable for our manuscript.
According to recommendations of Reviewer #3 we revised our manuscript to meet all requirements:
- This has become especially relevant after the COVID-19 pandemic. It is known that damage to the microbiota during coronavirus infection occurs directly and indirectly, and intestinal dysbiosis and leaky gut syndrome are the key causes of post-COVID syndrome. This rich topic deserves a separate review, and it would be great if the professional team of the authors would consider writing such an article. This review contains a large number of references, is excellently structured, easy to read and contains a wealth of information useful to both scientists and medical practitioners. For ease of perception, the material is summarized in several tables and one diagram. The article may be published in present form.
We had to restructured the whole manuscript according to Reviewer’s #1 requirements. Thank you very much for pointing out another interesting topic. We will gladly undertake the preparation of a manuscript regarding probiotics in post-COVID syndrome therapy.
Round 2
Reviewer 1 Report
The authors addressed the suggestions and answered the questions. The paper was restructured, the repeated data removed, thus, the whole manuscript has been improved.
I have no further comments.